# Conceptualization of Roma in Policy Documents Related to Social Inclusion and Health in the Czech Republic

**DOI:** 10.3390/ijerph17217739

**Published:** 2020-10-22

**Authors:** Lenka Slepickova, Daniela Filakovska Bobakova

**Affiliations:** 1Olomouc University Social Health Institute, Theological Faculty, Palacky University Olomouc, 771 11 Olomouc, Czech Republic; daniela.filakovska@upjs.sk; 2Department of Health Psychology and Research Methodology, Faculty of Medicine, Pavol Jozef Safarik University, 040 01 Kosice, Slovakia

**Keywords:** conceptualization of Roma, policy documents, social inclusion, health, Czech Republic

## Abstract

In the Czech Republic, a number of strategy papers and policy documents are guiding the direction of Roma inclusion, including in the area of health. The conceptualization of Roma and how mainstream political and public discourse operate with the term “Roma” contribute to a mistakenly homogenous and harmful image of Roma that conforms to negative stereotypes. The aim of our study was to examine the conceptualization of Roma in policy documents related to social inclusion and health in the Czech Republic. Relevant political, strategic and project documents were selected for analysis. Emphasis is placed in them on individual responsibility in relation to health, while structural conditions and discrimination are mentioned less often. Roma are described in relation to health primarily as people who should be educated. More emphasis is placed on the economic benefits of eliminating health inequalities than on citizens’ rights and the importance of inclusion. When “participation” or “empowerment” is mentioned, it is done vaguely, usually in addition to references to completely non-participatory practices. The majority is the primary actor in the field of eliminating health inequalities, as it defines the “path” that Roma need to be shown or determines what is needed to “stimulate” citizens. Although the political discourse concerning Roma has shifted more towards human rights, equity and combating discrimination in the Czech Republic, subtle forms of anti-Gypsyism still seem to be present.

## 1. Introduction

Indiscreet use of the term “Roma” to label people of distinct socioeconomic position from various subgroups (Roma, Sinti, Gypsies, Travelers and other) leads to a perception of supposed homogeneity of Roma associated with a number of negative life circumstances, such as social exclusion, marginalization, vulnerability, poverty and dependency on social benefits [1]. Roma constitute the largest ethnic minority in Europe, and it is well documented that many Roma live in socially excluded or marginalized communities and face discrimination, poverty, deprivation and inequities in all aspects of everyday life [2,3]. In recent decades, a number of relevant European and international organizations, including political and non-governmental bodies, have focused on human and minority rights advocacy on behalf of Roma and have encouraged national governments to address inequities faced by Roma in four key areas: education, employment, housing and health [3]. Sedulously pointing out the historical and political context as well as the structural factors and societal discourse that form the basis of the current unfavorable situation of many Roma living in social exclusion has led to a gradual change in political discourse concerning Roma. This discourse has shifted from crime prevention, population regulation and assimilation since before the early 1990s towards human rights, equity and combating discrimination [4]. Although this is viewed as a positive change, debates on the categorization and conceptualization of Roma and the consequences of how mainstream political and public discourse (including policies of the Council of Europe, Roma Integration Strategies in the EU, etc.) operate with the term “Roma” is ongoing. Social scientists argue that it is convenient but misleading to place various groups of people with various social positions in society under a single roof and that it creates a falsely homogenous and harmful image of Roma that conforms to negative stereotypes [1].

Many European states have adopted National Roma Integration Strategies focused on the above-mentioned key areas, and policy, legal and funding instruments have been aligned and mobilized since 2011 when the European Commission called for such action [5]. In recent years, growing attention has also been paid to fighting anti-Gypsyism as a separate priority area for Roma integration strategies at both the EU and national levels [3]. Anti-Gypsyism, defined as specific racism towards Roma which includes not only expressions and practices such as violence, hate speech, exploitation, stigmatization and the most blatant kind of discrimination but also various implicit and hidden manifestations [6,7], is perceived as an obstacle preventing the fair and equitable access of Roma to education, employment, housing and health [3]. According to Fundamental Rights Agency, member states have failed to address anti-Gypsyism due to a “shocking lack of action”; moreover, structural and institutional discrimination and anti-Gypsyism embedded in structures, organizations and institutions creates an additional barrier for Roma inclusion [3]. Although Roma Integration Strategies adopt and promote a progressive/liberal agenda that aligns with the strategic goals of the EU, subtle forms of racism can still be identified in strategy documents concerning Roma inclusion, which uncritically reproduce problematic assumptions about Roma [8].

The Czech Republic is among the EU Member States with the largest Roma communities and facing the most acute challenges [9]. The Roma population in the Czech Republic is estimated to be 240,300 (2.2% of the population) [10], and about 50% of Roma live in social exclusion [11]. Roma are underrepresented when it comes to political participation and representation. According to the Civil Society Monitoring Report on the Implementation of the National Roma Integration Strategies, the representation of Roma men and women in the Czech Parliament after elections in 2017 was zero, and ten Roma men and women were working in the state administration. Moreover, the Czech political scene is marked by the presence of openly anti-Roma, racist, xenophobic parties, and anti-Roma or racist speech appears abundantly from standard, mainstream parties and politicians [9].

Regarding the health care system of the Czech Republic, its overall performance may be crucial for determining how it acts towards Roma. Overall, in past decades, the health care system has been successful in bettering its performance in various routinely monitored indicators, such as health expenditures per person, effectiveness, accessibility and resilience. There seem to be only marginal unmet needs for medical care, which can be attributed to universal population coverage, a broad benefits basket, low-cost sharing and a high density of providers [12]. However, despite many improvements, the Czech health care system remains below the EU average in many other indicators [12]. The Ministry of Health itself acknowledges significant disparities in the fulfilment of the objectives of the individual action programs of Health 2020 and the largely delayed implementation of the strategy due to insufficient financial coverage for the planned activities [13]. The health status of the Czech population is characterized by substantial inequalities by education and income [12,14]. Vast regional disparities are characteristic of the overall distribution of primary care providers, who play a vital role in health promotion and prevention and as the first contact point in the health care system across the country [12]. The planning of health care services is aggravated by fragmented responsibilities in planning among several stakeholders (health insurance funds, self-governing regions, the Ministry of Health, the Ministry of Social Affairs, etc.). This causes the higher health needs in some Czech regions to not be properly accounted for in planning [12].

As the evidence suggests, the health status of Roma is worse than that of the non-Roma population in the Czech Republic. Roma life expectancy is about 10–15 years less than the majority population and there is a significantly higher prevalence of chronic illnesses in the Roma population. The infant mortality rate is also higher among Roma [15]. The Czech Roma population smokes significantly more than the non-Roma population and only low numbers of Roma (adults and/or children) undertake physical activities to stay healthy. The Czech Roma population typically has a poor diet, with less than the recommended daily intake of vegetables, fruit and dairy products, and above the recommended intake of meat, sweets and processed food [15]. The reasons for the worse health status of Roma are complex and cannot be easily identified merely as an “unhealthy lifestyle” or “risky health behaviour” or seen as individual characteristics, since health-related behaviour is rather the consequence of exclusion, worse living conditions (especially housing), poverty and the lower social, economic and cultural capital of the Roma population [15,16]. What is more, the European Roma population generally faces disproportionate barriers to accessing health services, and the Czech Roma population is no exception. This situation is caused by several factors: living in remote areas without necessary health and transport infrastructure, a lack of financial sources or insurance, as well as discriminatory attitudes [15,17,18]. The health needs of the Roma population are often invisible because of the absence of research but also due to the absence of advocacy on their behalf, as Koupilová et al. suggest [19].

The “Roma Integration Strategy of the Czech Republic” for the periods 2010–2013 and 2014–2020 followed the recommendations of the European Commission and aimed to cover four key areas as well as the fight against discrimination [20,21]. In parallel with the creation and implementation of the Action Plans of the Strategy, other actors on local and non-governmental levels operating in the field of Roma inclusion and advocacy for Roma rights have been engaged and have formulated their own agenda. As a result of multiple initiatives on various levels, a number of strategy papers and policy documents are now guiding the direction of Roma inclusion, including in the area of health, which seems to be underdeveloped and understudied in the Czech Republic. Thus, the aim of our study was to examine the conceptualization of Roma in policy documents related to social inclusion and health in the Czech Republic, to determine to what extent Roma are assigned an active or passive position in relation to their own health and to identify hidden problematic assumptions and/or depictions about Roma.

## 2. Materials and Methods

This study was carried out within the Technology Agency of the Czech Republic funded ROMZAVIP Project (TL02000164), which aimed to innovate the existing methodologies and procedures concerning the integration of Roma people living in social exclusion in the area of health, focusing particularly on health care accessibility. The study was approved by the Ethical Committee of the Social Health Institute, at Palacky University Olomouc, under reference number 2018/09, in December 2019.

### 2.1. Design

The project activities also included a qualitative analysis of documents related to social inclusion and health. The analysis aimed to describe how the analyzed strategic documents represent the Roma in relation to their own health, to what extent they are assigned an active or passive position in relation to health, or to reveal hidden elements of assimilation, as well as other conflicting themes and inconsistencies.

### 2.2. Sample

Political, strategic and project documents dealing with health, the integration of the socially excluded or the Roma in general were selected for analysis, and their validity dates back to the period from 2015 to the present. Selected documents have been developed both for the whole country and for different regions. The list of the documents for analysis was consulted and finalized in cooperation with several stakeholders working in the field of social inclusion or health. Apart from the recommendations of the stakeholders, the selection of documents was led by the principle of theoretical saturation, i.e., it ended at the moment when no new or relevant data seemed to emerge by analyzing new documents [22].

A total of 929 pages of text were analyzed. For the list of documents included in the analyses, see Table 1.

### 2.3. Procedure

The analysis was guided by the following main research questions: How are the Roma presented in the documents as actors in relation to health? What role is ascribed to them? How much is their own participation expected? The original research questions were, in accordance with the inductive logic of qualitative analysis, rather broad [22] and the analysis became more focused with the first topics and categories emerging in the data, as described below.

### 2.4. Statistical Analysis and Reporting

The qualitative analysis was done in a purely inductive way, without pre-prepared categories, using the NVivo Pro 12 software. Significant text passages from the documents (193 excerpts in all) were then fitted with codes, the original number (43) of which was reduced to 30 in the subsequent phase, based on the frequency of use and the relationship to the research questions. For the analysis, the most important particular codes were then assigned into broader categories which reflect the individual subchapters of the analysis below. Proceeding from the particular to the more general is one of the key principles of the qualitative analysis [22] and enables us to present our findings as contextualized.

## 3. Results

In the course of repeated reading, fragmentation and reduction of data and their recompiling and a search for meaningful patterns, six main themes were defined which are characteristic for the way Roma are presented in relation to their health in the analyzed documents. This not only involved the main themes mentioned in the documents in this context, but also the ways of arguing, reasoning and presenting individual opinions. Table 2 presents the names of these categories/domains and codes (as the tools for reducing, sorting and categorizing data) that are included in them.

### 3.1. Why Address Health Inequalities at All? “Leave No One Behind”

Many documents testified to the existence of health inequalities in the Czech Republic, even though these are not always related to the Roma or even more generally to people living in social exclusion. Some documents speak about “population risk groups”, about the “socially weak” or about “vulnerable groups of the population”, “the population of structurally affected regions” or “risk segments of the population”, or about problems “in remote localities”. The document “Health 2020” barely deals with inequalities in health at all; based on reading it, the main risks to health in the Czech Republic seem to be infectious diseases, exceptional situations, a lack of health monitoring or insufficient vaccination coverage. Only the appendices with data on the health status of the population of the Czech Republic, and in particular the Czech Republic’s lagging behind other countries, point to specific problems.

Some of the documents addressed the overlapping of the categories of socially excluded and Roma, such as, for example, the “Strategy for Integration of the Roma Minority in the Central Bohemian Region”, which states that it focuses on the portion of the Roma population that is also affected by social exclusion. According to this document, many Roma are commonly involved in society and do not need integration. Socioeconomic exclusion is more important than ethnicity. According to the “Strategy for Integration of Socially Excluded Localities in the Vysočina Region”, in the case of the Roma, ethnic discrimination is added to the problems posed by social exclusion. The fact that it is ethnicity that stands behind the discrimination and exclusion has not been addressed very often.

Those documents that dealt explicitly with Roma health and health inequalities demonstrated this with citations from various studies which focused on comparing various indicators (life expectancy, number of smokers, eating habits, prevalence of different diseases, including mental health) of the health status of Roma and the majority population. What are the arguments of these documents for the battle against health inequalities? For the sake of clarity, we describe these arguments as “ideological”, “financial” and “utilitarian”, and they are often combined in individual documents.

The first of the arguments is ideological; it is possible to say “inclusive”, expressed in the principle cited in the document “Health 2030”: “Leave no one behind” (p. 16). The document “Europe 2020” emphasizes that there is a need to ensure that “everyone can benefit from growth” (p. 19). This should be achieved by combating poverty and social exclusion and by “reducing health inequalities” (p. 19). The document “Health, Health Inequalities” refers to the “Final Report” of the WHO Commission on Social Determinants of Health 2012, which states that “systematic health inequalities are perceived as unwarranted and unjust and need to be resolved with the participation of all sectors of society, including government departments” (no pagination).

Another commonly appearing argument is the financial one. In the document “Health 2020”, this is specifically the argument for health care costs. According to this document, these costs can be “substantially reduced” by “implementing known and evidence-based prevention methods” (p. 14). It is necessary to strengthen preventive public health since managing problems in health care facilities is “very costly and economically unsustainable” (p. 14). The promise lies in the fact that “the desired outcome—the good health of our population—will bring about benefits for all sectors of society” (p. 7), will save health care costs and will also have other economic benefits. In contrast, “should we fail to fully utilize the potential of effective disease prevention and health promotion, the costs of health care will inevitably keep rising, which, in the long term, is not sustainable” (p. 6).

The document of the Agency for Social Inclusion, “Health, Health Inequalities in Health Care” also estimates the financial losses caused by health inequalities at 78 billion per year (1.5% of the Czech Republic’s GDP). “Of this, 26 billion represents direct losses in medical costs, to which the documented forced postponement of the solution to the health problem further contributes. In consequence of postponement of a health problem, the numbers of work disabled, premature mortality, amount of social benefits, etc. are increased, which contribute to the remaining 52 billion in indirect losses” (no pagination). Therefore, it is necessary to work on a “more effective solution for prevention of social exclusion and access to health care” (no pagination).

Both types of arguments—ideological and financial—are contained in the document “Health 2030”, which states that “such a strategy is an expression of responsibility and concern for the future. This is about respecting basic human values, to which human health belongs” (p. 6). At the same time, it emphasizes that “the health status of the population is expressed in the measure of employment or the burden on the health and social care system and is thus an important determinant in the national economy” (p. 9). Therefore, it is necessary to try for effective prevention, because even though a whole range of health problems in the population is due to demographic development (ageing of the population), many of them are also associated with a poor or unhealthy lifestyle. “Disease prevention in particular thus represents a key tool for improving the health of the Czech population and improving the efficiency of the health care system, thereby increasing the employment of the population in the national economy” (p. 55).

The document “Strategy for Integration of the Roma Community in the Moravian–Silesian Region” specifically relates inequalities to Roma living in social exclusion and emphasizes health as an important prerequisite for social integration of Roma, as it conditions their self-sufficiency and participation in the labor market. It is therefore important, according to the document, to deal with Roma living in social exclusion, for whom access to health care is at risk. We call this argument (the better health of the Roma ensures their greater participation in the labor market) the utilitarian argument.

### 3.2. Health as an Individual Choice: “Let’s Live a Healthy Life”

Most of the documents link the poorer health of the Roma (described through research comparing various parameters of the health of the Roma and the majority population) with their lifestyle, “risk behaviour” and individual choices (“underestimating prevention”, etc.). Moreover, those documents that list the structural causes of poorer Roma health often propose measures which attempt to correct individuals through “education” (sometimes even education aimed directly at children) and do not address at all the situation’s more general context. Health is perceived as a matter of individual responsibility and choice, and the Roma as those who must be “educated” and for whom “the responsibility of the individual for his/her own health status, to the extent that he/she can influence it through his/her actions” should be emphasized (“Health 2030”, p. 11). According to the document “Strategy for Integration of Socially Excluded Localities in the Vysočina Region”, individual responsibility also extends to a person’s integration, that is, it is up to the individual to achieve what the majority has: “Thus, in an effective integration process, individuals who take an interest and who are able to take responsibility for their lives, who are anchored in a job, a wage, in providing for their children, education, work and a professional perspective, gain opportunities” (p. 7).

The document “Health Promotion in Excluded Localities” assumes that the Roma lack the courage to resist the pressure of the environment and thus to free themselves from their “non-standard” way of life. It is necessary for the majority to show them “that there are other ways”. The Roma are, according to this document, motivated in their life strategies by the effort to “surround themselves with comfort and luxury”.

“We don’t have to try to help them at all costs; we only have to watch, listen and think about the causes of their behaviour and their non-standard way of life. To show them that there are other ways, but that courage is needed to follow them. The residents of these localities, like most of the majority society from which they are excluded, prefer conformity. They want to resemble the majority, to be surrounded by comfort and luxury. This means hamburgers, cola, cigarettes, chips, sweets, Adidas shoes, mobile phones and televisions for them. The opposite of courage is not cowardice but comfort” (p. 24).

“The adverse health status of the Roma is a consequence of their way of life and not of their ethnicity”, writes the author of the document “Health Promotion in Excluded Localities” (p. 4). The author based the statement on the fact that the poor around the world display similar indicators of poorer health. However, elsewhere in the same document, the author considers the structural causes of the poorer health among the Roma: “Health inequalities in the population of socially excluded localities are caused by lower education and worse socioeconomic conditions. People living here have limited access to information, which is often given to them in a completely incomprehensible or complicated form. People with low incomes, without education and dependent on social benefits often don’t consider a healthy lifestyle as a priority. An inappropriate diet, a lack of exercise, the risk of addiction and impaired access to health care are problems that can threaten people living in socially excluded locations more than in the majority population. The lack of proper nutrition and a generally unhealthy lifestyle in children can have a negative effect on school success and later integration into society” (p. 7).

The naming of various aspects of the Roma lifestyle as the main causes of their poorer health status is very common in the documents.

As an example, the “Strategy for Integration of Socially Excluded Localities in the Vysočina Region”, mentions the fact that people living in social exclusion focus only on dealing with basic living needs as an “irresponsible approach to their own health” (p. 49); it also talks about risk behaviour and a “bad” lifestyle (p. 49).

The “Strategy for Roma Integration in the South Moravian Region” describes, as part of individual attitudes, which are one of the causes of poorer Roma health status (in addition to institutional barriers), the “neglect of preventive examinations (gynecology, dentistry), non-adherence to treatment of diagnosed diseases, an inappropriate lifestyle (particularly smoking during pregnancy and motherhood, unhealthy lifestyle), stress, resignation, apathy” (p. 15).

A poor diet is one of the often described aspects of the Roma lifestyle: “Their diet is based on the consumption of sugars and polysaccharides and meat; it lacks vegetables, fruits and fish and there is insufficient consumption of dairy products” (“Health Promotion in Excluded Localities”, p. 4). It is expected that women will be more sensitive to preventive health-related activities, not with health as a motivation but because of their appearance: “Nutrition seems to be the most enforceable, if only because it is mostly women—mothers—who look for health advisory services and listen. Roma women, too, want to be slim and healthy, the ideal of fullness is rather a widespread stereotype about the Roma” (p. 19).

Another problem observed and also dealt with extensively in the document “Health Promotion in Excluded Localities” is the lack of physical activity (p. 20). The document describes the causes of the lack of physical activity among Roma as structural (a lack of financing for organized activities, for household and childcare for Roma women, and also the fact that Roma are “not welcome” (p. 20) in sporting areas, such as swimming pools or gyms). At the same time, an appropriate solution, in the author’s opinion, is to “take into consideration the possibilities of the community, family or individual” (p. 20) and “to recommend participation in activities for the general public with caution and focus more on the importance of walking or running” (p. 20).

Likewise, avoiding visits to a doctor is also described as part of the Roma lifestyle that leads to their poorer health. Smoking and drug addiction are often presented in such a way. The “Strategy for Roma Integration in the South Moravian Region”, similar to a number of other documents, sees correction of the situation as a matter of individual decisions by Roma, who will be encouraged to do so “in the scope of education” (p. 15). This should emphasize “that these inhabitants have their own doctor and visit him/her regularly for the purpose of preventive examinations, do not live in a harmful environment (hygienically unhealthy flats), do not unnecessarily worsen their social situation as a consequence of illness or disability (disability pensions, or other benefits)”(p. 15). Having a doctor and not living in a harmful environment is thus again perceived as a matter of individual decision, which education should help to address.

The “Strategy for Health Promotion and Health Services Development in the Ústí nad Labem Region” also presents the health status of Roma as poorer than in the majority population, but it proposes a solution that targets almost exclusively individual behaviour, such as “the elimination of risk behaviour”, “education and health promotion programmes”, “stress management and mental health”, “proper nutrition and eating habits” as well as “sufficient physical activity” (p. 17).

The “Strategy for Integration of the Roma Community in the Central Bohemian Region” also emphasizes the importance of “educational activities among Roma citizens (lectures on the health sciences, a healthy lifestyle, smoking and alcohol, preventive examinations, vaccinations, child care, gynaecology and planned parenthood, dentistry, etc.)” (p. 32). The best way to remedy the situation in the view of many documents is education at the individual level, including education for children, described in the document “Health Promotion in Excluded Localities”; thus, “A programme was created for increasing the level of knowledge of primary and secondary school pupils regarding infectious diseases and adopting practical habits to prevent them. The pupils will learn about different types of microorganisms, paths of transmission and the spreading of diseases, as well as basic hygienic measures in the spread of infections. Through illustrative and entertaining activities, pupils will become acquainted with the principles of good hygiene and protection against infectious diseases and will thus become aware of the importance of personal hygiene to prevent infectious diseases and the importance of vaccination” (p. 14).

The frequently mentioned idea of health literacy plays a specific role here. This is discussed in the “Strategy for Roma Integration” and in the documents “Health 2020”, “Health 2030” and the “Report on the Status of the Roma Minority”. Health literacy is defined, for example, as “the ability to obtain, interpret and comprehend basic health information and services and the ability to use such information and services for better health” (“Health 2020”, p. 23). The documents contain the assumption that “a systematic increasing of their health literacy (note: the health literacy of the Roma) is an important step towards achieving Roma integration or eliminating health inequalities” (“Health 2030”, p. 6), “strengthening health” (“Health 2030”, p. 60) and “strengthening the role of individuals and communities in caring for their own health” (“Health 2020”, p. 6). Only a person with adequate health literacy can care for their health and, in doing so, eliminate the problem of poorer health in a certain portion of the population.

Studies, such as the one conducted in the South Bohemian Region, point to poorer health literacy among Roma (62.5% of Roma had insufficient health literacy compared to 49% of respondents overall) (“Report on the Status of the Roma Minority for 2018”) and the association between health literacy and socioeconomic status, family status and employment (“Report on the Status of the Roma Minority for 2018”). Insufficient health literacy is likewise associated with risk factors and patterns of risk behaviour for health (“Health 2030”).

The document “Strategy for Roma Integration in the South Moravian Region” also refers to health literacy problems, without naming them; however, it states—as one of the reasons for the poorer health status of the Roma—“the inability of some Roma to receive and analyse the information available (relating to lifestyle, the demands and rights associated with the use of public services, etc.) and the information provided by doctors and health care personnel (often due to language barriers and communication skills; the users do not understand the information provided, thus many are often unable to adapt their treatment behaviour to requirements, etc.)” (p. 15). The “Strategy for Social Integration” further states that the Roma have “low awareness of the rights and obligations of a public health insurer or patient” (p. 45).

Again, the remedy consists of education and activities that raise awareness, which is particularly important for so-called “at-risk groups” (“Health 2030”, p. 29)—an example is a project by the Municipal Charity of České Budějovice, which worked to increase the health literacy of Roma families, or so-called “health days”, such as the one in Nový Bor in 2018, which took place under the title “Let’s live healthily” (“Report on the Status of the Roma Minority for 2018”).

The document “Strategy to Combat Social Exclusion” considers on page 6 why views on social exclusion in the Czech Republic are dominated by the view called a “moral approach to the culture of poverty”, which sees the cause of poverty and exclusion in the characteristics and behaviour of the poor themselves, not in the failure of the society, which we have illustrated above using the example of many of the cited documents. According to the document, the “specific dimension of social exclusion in the Czech Republic, which is ethnicity” has a significant influence on this. In addition to the individual causes of the poorer health status of the Roma, these documents also mention, although less often, barriers on the part of the health system, as the following part of the analysis shows.

### 3.3. Structural Causes of the Poorer Health Status of Roma: “An Unsuitable Environment for Spreading any Education”

Aside from the causes of health inequalities on the individual level, a number of documents examine the structural causes of the poorer health status of Roma.

Many documents mention the issue of health care access but often only in general and not in relation to the Roma. For example, “Health 2030” lists several factors influencing access to health care: “health insurance coverage, the care package (depth of coverage), affordability of care (co-payments, cost-sharing) or the availability of care (health professionals, distance, waiting times)” (p. 18). It then considers the support of health care access for “vulnerable groups” (p. 18), such as the socially excluded, homeless, dependent, etc., as the goal. It points out the recent decline in health care access in the Czech Republic due to the declining number of staff, particularly in remote locations.

Several documents address the accessibility of health care, specifically as one of the structural causes of the poorer health of the Roma. The “Strategy for Integration of the Roma Community in the Moravian–Silesian Region” deals with the issue of health care access, which relates mainly to those Roma living in excluded localities. According to the document, the causes for this situation come from the community itself (a lack of information and education, low interest) as well as systemic causes, such as “the social exclusion itself of the inhabitants of excluded localities and problems related to the provision of specialized health services, e.g., dental care, etc. The knowledge emanating from the practice of field social workers shows that problems occur with health care provision in connection with prejudice and discrimination” (p. 29).

The “Report on the Status of the Roma Minority for 2018” similarly points out that the shortage of doctors in the Czech Republic affects the Roma more to the extent that Roma patients are rejected. The “Strategy for Integration of the Roma Community in the Central Bohemian Region for 2017—2021” also describes on p. 32, as part of the Roma lifestyle which leads to their poorer health, purely structural circumstances: “disadvantageous living conditions (poor quality water, mould, etc.)”, “the performing of physically demanding work long-term, sometimes even in an unhealthy environment” or “stress caused by everyday worries”.

The “Strategy for Integration of Socially Excluded Localities in the Vysočina Region” names a series of structural causes of health inequalities. It proposes education as the primary type of solution—“education related to health prevention” (p. 50)—but it also targets systemic changes and not just education at the individual level: it should consist of support for the education of health care field workers, support for fieldwork aimed at minimizing risk behaviour (basic hygienic habits), support for cooperation between municipalities, schools, hygiene stations, doctors and other subjects as well as support for the creation of social and health assistant jobs. Likewise, the “Strategy for Roma Integration in the South Moravian Region” also names individual attitudes and habits as the cause of the poorer health status of Roma, but it also refers to “institutional barriers in the use of the health and social care system associated with a disability” (p. 14).

The “Strategy for Integration of Socially Excluded Localities in the Vysočina Region” mentions the problem of the availability of services, but also considers this to be unconfirmed, referring to the guarantee of health care availability by law: “The experience of field workers shows that the health situation in socially excluded localities is directly proportional to the quality of housing and the availability of services. (…) We do not, however, register any official or unofficial suggestions that would indicate the inaccessibility of necessary health care. Health care availability is guaranteed by law. According to Act No. 372/2011 Coll. on Health Services” (p. 46).

The unavailability of health care, according to the documents, is—in part—caused by discrimination (doctors refusing to register Roma patients), but also by physical inaccessibility, which can be insurmountable for Roma patients and which is reinforced by the lack of doctors in certain specializations in certain regions of the Czech Republic. One document from the Agency for Social Inclusion (“Health, Inequalities in Health Care”) is testimony to this issue: “Even in the present, a trip to a paediatrician is essentially an all-day affair for the citizens of municipalities such as Bulovka, where a mother is often forced to take other, healthy children to the doctor because she would not be able to get home by the closing time of the nursery school due to difficult location and time availability” (no pagination). This situation affects both the patients and the doctors who care for them, as they often exceed the number of registered patients allowed and work at the expense of their free time (“Health, Inequalities in Health Care”).

Although many documents do mention discrimination, it is often in a very general way (as, for example, one feature of the life of the people living in social exclusion), or in relation to groups of people other than Roma (discrimination against women, the elderly or the disabled), or they do not name it directly. We can find specific mentions of discrimination in relation to the availability of health care and the Roma or people living in social exclusion in the document of the Agency for Social Inclusion (“Health, Inequalities in Health Care”). The Agency acts in the battle against discrimination “through building civil law awareness of people living in social exclusion in the area of the rights and obligations of the insured. This includes specific procedures when refusing treatment and registering with a primary care provider” (no pagination). The document also describes indirect discrimination, “when some Roma mothers, in particular, have encountered or still encounter the fact that a dentist, paediatrician or gynaecologist repeatedly refuses to register them. Indirect discrimination consists in the fact that for a Roma mother, the doctor is almost always filled up” (no pagination). This document also describes a specific situation in the Frýdlant Region, where a married couple, both doctors, opened a surgery for Roma clients who find it difficult to register with general practitioners, and also mentions the unavailability of mammography for women who are at risk of poverty and social exclusion “as a consequence of economic and social barriers” (no pagination).

The document “Strategy for Roma Integration in the South Moravian Region” also addresses the problems of Roma patients in registering with doctors “associated relatively frequently with inappropriate behaviour” (p. 15). It also indicates recent changes in the obligations associated with the payment of health insurance but also in the emergence of a claim on a disability pension as the cause of the poorer health status of the Roma. Many Roma have debts with their health insurer or do not receive a disability pension because they do not meet the statutory insurance period. However, the document sees the main cause of the poorer health status of the Roma in exclusion itself, that is, in the “social and spatial isolation of the target group from the majority society. There is a complete lack of reference groups coming from other social (i.e., economic, professional, educational) groups which would confront the target group with other life strategies, levels and styles” (p. 15).

In connection with health literacy, the thematization of which was described in the analyzed documents in the previous subchapter, references are made to systemic conditions for its improvement, such as the availability of “quality and comprehensible information with the use of people-focused information systems, such as the media, eHealth, web or consumer information for consumer products” (“Health 2020”, p. 23), or, in other words, “the creation and implementation of suitable communication channels, such as, for example, the launching of the National Health Information Portal or improving the communication skills of health professionals through activities aimed at processing professional information into generally understandable forms” (“Health 2030”, p. 61).

For example, the document “Health Promotion in Excluded Localities” describes, in a relatively dramatic way, the causes of drug addiction among young Roma (in relation to solvents) and the difficult environment for improvement (“spreading any awareness”), also in systemic terms: “The only purpose of their lives is to escape the reality that surrounds them. The ugliness of the buildings where there are hostels, no privacy, shared bathrooms, kitchens and toilets, material deprivation, the hostility of the surroundings, this is not an environment for spreading any education”. “They must pay heed to prevention! But how, when the future for several years is blurred for them by fog or even just a black hole in which there is nothing, nothing but darkness…” (p. 2).

The document “Strategy to Combat Social Exclusion” focuses on the structural solution of the present situation: “In order for people living in social exclusion to overcome their displacement from society, they need help from outside. The state should provide basic help and support to them through a set of tools. One of them is the appropriate setting and continuous correction of state policies, such that the structural barriers to the integration of people living in social exclusion into society are as small as possible. Another instrument is an appropriate intervention in regional and municipal policies with the aim of creating conditions for implementing social inclusion at the local level; further effective support for services intended for direct assistance to people living in social exclusion; no less important is the activity of creating an anti-discriminatory social climate and appropriate intervention in public discourse. Last but not least, activation of the people living in social exclusion themselves and ensuring their active participation in changing their life situation is also fundamental” (p. 6).

The “Strategy for Roma Integration to 2020” emphasizes that the Ministry of Health and health insurance companies are the guarantor of access to health care—they should pass measures to prevent discriminatory practices. “Great emphasis should be placed on a professional approach to clients from different cultural backgrounds. Despite the validity of the Anti-Discrimination Act, it is clear that the inhabitants of socially excluded localities, in particular, do not take advantage of the opportunities given by the mentioned legal standard, unless they are in contact with an organization that deals with the issue and can provide them with professional support” (p. 67). It proposes incorporating this topic into the education of health professionals and other helping professions and officials.

The “Strategy for Health Promotion and Health Services Development in the Ústí nad Labem Region” describes the necessity for pressure on health insurance companies aimed at “strengthening the insufficient capacities identified within individual health services” (p. 18) and also on the need to support “balanced development of health services infrastructure from EU structural funds” (p. 18), but without a specific reference to the Roma.

The documents “Health 2030” or the “Report on the Status of the Roma Minority for 2018” emphasize the cohesion of structural and individual causes of the poorer health status of Roma. “The impact of substance abuse, other forms of risk behaviour and socioeconomic determinants of health (poverty, unemployment, loss of social cohesion, low access to health care) combine and potentiate each other” (“Health 2030” (p. 30)). The “Report on the Status of the Roma Minority for 2018” presents factors ranging from a poor lifestyle to “inappropriate approach of doctors to the Roma minority” as causes for the poorer health status of Roma (p. 40).

The “Strategy for Integration of the Roma Community in the Central Bohemian Region” similarly emphasizes on page 2 that integration depends on both people living in social exclusion and in the society itself: “integration is a two-way process and that some signs of risk of social exclusion, for example, difficult access to services or social networks are not only caused by ‘insufficiency’ on the side of integrated persons but also the closed nature on the part of those being integrated. In consequence, the situation frequently occurs in which a person is prevented from fully integrating by society itself only based on prejudices”.

### 3.4. Education of Health Care Professionals on the Specifics of the Roma Population: “Ignorance Causes Misunderstanding”

Some documents focus on the need for specific forms of education, namely among health care professionals and other professionals who work with the Roma. They emphasize that Roma as clients or patients have their own specific characteristics, which the people who work with them should know about and which they should adapt to. This usually involves a necessary adaptation to cultural differences and traditions (who has authority in a Roma community?), the social experience of this group (their distrust of institutions) or lower health literacy and lower education. Sometimes the involvement of Roma as assistants or mediators in the provision of health and other services is described as an appropriate solution.

For example, the “Strategy for Roma Integration” describes the Roma as “people with different social and cultural experiences” (p. 67) and promotes the “systematic and standardized education of health care professionals” (p. 67) for working with this ethnic minority. Health care personnel should be “qualified and culturally sensitive” (p. 67). According to this document, it is important to emphasize “the helpfulness of health care personnel in the provision of health care and for the adequacy of information provided to minorities and patients coming from other cultural backgrounds in the interest of creating mutual trust and respect as a prerequisite for quality health care” (p. 67).

This document also focuses on the specifics of Roma clients in the area of social work: “For example, Roma clients may more often show distrust towards social workers of the majority society (especially in reaction to previous bad experiences), distrust towards other institutions of the majority society, a need to overcome certain traditional attitudes, such as the lower importance of education or stereotypes about the role and social status of women, undervaluing the importance of healthy and adequate nutrition for maintaining health”. It proposes that “Roma who know these specifics and further have in the intention of the Social Services Act the opportunity to raise their knowledge and competencies through further education work in teams of social service providers”.

The method of providing Roma with health-related information is addressed in detail in the “Health Promotion in Excluded Localities”, which emphasizes that “when sharing professional knowledge, the information needs to be reduced to comprehensible, clear and unambiguous facts. Use a minimum of foreign words, if they can be replaced without changing the meaning. There is no need to use Romani because many Roma do not know it or they use it only to a limited extent; furthermore, the Romani language differs in different areas. Czech is decidedly not an obstacle, nor does it cause feelings of discrimination” (p. 21). According to this document, honest, positive and supportive non-verbal communication, a non-critical approach, patience and optimism among health care professionals are a necessity for “achieving the aims of all interventions”; thus, they are more a means to an end, which is the support of health. “It is likewise important to be aware of the complexity of living conditions and their interrelationships—the financial situation, discriminatory moods in society, the form and quality of housing—but even so to not be overwhelmed by the negative feeling that health promotion cannot be done because it is not a priority for the target population” (p. 21).

The document also recommends involving intermediaries, ideally other Roma, in health promotion interventions, because the trust of the Roma for the medical profession is not automatic, especially when the front-line worker is young or a woman. The document also contains a range specific advice and recommendations relating to the timing of interventions (this is not ideal in situations of existential threat), but also the appropriate rhetoric, which should primarily be used “to point out the benefits of change”, even though it is sometimes necessary to target their fear, but to do so, of course, “sensitively and after consideration of the preferences of the target group” (p. 24), for example, “in the spirit of ‘and they said on television…’, which is a magical universal synonym for the truth. This applies three-fold in excluded areas, where television is the only source of information” (p. 24). Therefore, in this document, adaptation to the specifics of the Roma population is perceived purely pragmatically, as an important instrument for promoting what the majority wants to achieve in the field of health.

### 3.5. Participation Primarily as Better Care of Individual Health: “The Need to Stimulate Citizens towards Responsibility”

“Empowering the patient” or a “partnership” is not dealt with very often in the documents, and if it is, only in a very general and vague way, and proposals for fully non-participatory approaches and interventions are found linked to declarations of equality, participation and partnership. As an example, the “Health 2020” document states that “The aim is to create a sustainable health system based on quality, availability and equality of people who become partners in attaining better health for all” (p. 6).

More specifically, this document mentions this as a task for health care professionals, which professional preparation should focus on (among other things). Aside from strengthening the role of patients, these professionals should focus on improving “self-care by patients” (p. 25). The document further emphasizes the necessity for the “participation” (p. 7) of communities and individuals for improving the health status of the population and the necessity for the “active engagement of citizens” (p. 26) and hearing the “voice of civil society, including individuals and patient organizations, youth organizations and senior citizens” (p. 20). These should point out circumstances that are harmful to health (dangerous lifestyles, defective merchandise and shortcomings in the quality and provision of health care) and be a source of new ideas and thinking. The goal, according to this document, is “that independent and competent people understand the value of health and can accept their portion of shared responsibility not only for their own health but also for the health of those who need it” (p. 7). The document “Health 2030” similarly addresses the need to “stimulate citizens to make healthy choices” (p. 9). Rather than an emphasis on autonomy and participation, here we see an emphasis on education (“competent”, “understanding the value of health”) and accepting responsibility. The partnership is perceived here as better care for individual health and accepting responsibility, not a participation in decision making. The actor, in this case, is the majority, which decides what is needed to “stimulate” the citizens.

At the same time, one can also take responsibility for one’s own health through autonomous decision making in the wrong way, as mentioned in “Health 2030”, which points out the fact that vaccinations are declining in the Czech population as proof of “reduced health literacy and problematic assuming of responsibility for one’s own health (p. 31). The document “Health 2020” states in regard to the same topic that the Ministry is responsible for thorough vaccination (of interest is the mention in another part of the document that “infectious disease risks and coercive methods in health protection” dominate historically in the Czech Republic) (p. 15).

A Roma patient, as an autonomous individual who has the right to decide about his/her own health and then gets the necessary support, emerges, for example, from the “Health vs. Social Exclusion” document from the non-profit organization DROM. Cases and situations in which a health assistant can help are described here. It is evident from the language of the document that it is the client himself who is the actor in this situation. A health assistant is described as a person who can help (advise, accompany) in the event the client “needs” something (to register with a doctor, arrange a disability pension) or makes inquiries (“Or you just do not feel well, you do not know what’s wrong, and you’re afraid to go to the doctor”). “For those who want to change their lifestyle or eating habits, they will advise how” (no pagination). Roma patients are not seen primarily as a target of education related to individual health care, but as people who are endowed with rights that they may need help with. “Further, it will also help you with the resolving of hygienic problems in a flat, with the use of the right to preventive examinations paid for by health insurance and other similar issues that may trouble you” (no pagination).

In the area of social assistance, the document “Strategy for Roma Integration” generally calls for a more participatory approach and increased capacities and finances in social services. “It has also long been confirmed that social services with a high degree of participation in the lives of ‘clients’, including their involvement, especially through community work, are the most successful. (…) The trend should be to move the client from the field form of services to outpatient forms, which require greater responsibility and cooperation from the client” (p. 65).

## 4. Discussion

The aim of this study was to examine the conceptualization of Roma in policy documents related to social inclusion and health in the Czech Republic and to answer the following questions: How are the Roma presented in the documents as actors in relation to health? What role is ascribed to them? How much is their participation expected?

Roma are perceived in relation to their own health as having a “bad” lifestyle, as irresponsible and neglecting health care (usually in the sense of interactions with the official health system). They also lack information about their rights and obligations and have low health literacy. The solution to this situation, according to the studied documents, is “education”, the content of which is defined by the majority and the implementation of which is adapted to the specifics of their recipients so that it penetrates among them as best as possible (sometimes it is legitimate to scare them a little). Health promotion should contribute to helping people better “understand” the importance of health so that they take responsibility for their own health and make choices that benefit their own health (not living inappropriately, eating properly, not neglecting prevention). In the scope of health promotion, it is important that people who work with Roma know the specifics of their culture and are able to adapt to them. This adaptation is perceived in a purely pragmatic way, as an important instrument for promoting what the majority wants to achieve in the field of health. Our findings regarding this topic are in line with a similarly described situation in Slovakia [23]. This similarity can be attributed to the geographical and political closeness of the two national states, which have a shared history of a common state—Czechoslovakia. Although targeting health-endangering behaviours seem to be reasonable given that the Roma—particularly those living in social exclusion—do engage in such behaviours more frequently [15], the approach and rationale for actions as found in the policy documents seem to lack the awareness of the root causes of engaging in such behaviours, specifically in the population of socially excluded Roma [24] on one hand, or the theory of planned behaviour [25] on the other.

The documents also show that Roma often go unrecognized as actors in relation to health inequalities—they are not named and the topic of health inequalities in this group of the population is hidden behind terms such as “risk segment of the population” or “vulnerable groups”, which are designations that are perceived rather negatively (risk behaviour). As over-communication of ethnicity as well as under-communication of ethnicity may reinforce the unfavourable position of the Roma ethnic minority [1], optimizing the method of communication of particular issues in political discourse seems to be necessary.

Reducing health inequalities and investment in Roma is moreover justified economically and utilitarianly rather than ideologically: what will bring an improvement in the health status of the Roma to society is “increasing the employment of the population in the national economy” (“Health 2030”, p. 55). However, this unilateral view of the relationship between health and participation in the labor market is very simplistic. Health status, on the other hand, is affected by material conditions and the environment that an individual grows up in—including educational opportunities—and, at the same time, these conditions determine to a great extent a person’s opportunities on the labor market, as the theory of the social determinants of health demonstrates [26]. The way this issue is addressed in the examined documents places the responsibility on individuals who, nevertheless, have little or no control over the strongest determinants of health (structural determinants), as these can only be changed through social and economic policies and processes [26]. Although the “leave no one behind” rhetoric [27] occasionally appears in the documents, unfortunately, this is not adequately translated into the proposed solutions.

Documents that describe structural causes, aside from or in place of individual causes of the poorer health status of Roma, usually deal with health care accessibility—both physical, i.e., hampered by economic barriers and shortages of doctors in certain regions, and factual, i.e., prevented by discrimination and prejudice—or disadvantageous living conditions, mainly poor quality housing. Roma are thus perceived as people whose integration is hindered by the majority, in which reform should also be sought (be it through improving the financing of field services or putting pressure on insurance companies). Although the health care system has its special position as a component of social determinants of health framework [27], its role seems to be overestimated in the studied documents. Access to health care is frequently mentioned (and often as the only determinant of health), even if it is a factor that only partially explains health inequalities. Furthermore, this topic relates to those who are already trying to find a doctor because of a health condition (except for preventive examinations), so it only affects a certain and specifically defined issue related to health status, which is preceded by many other determinants [28]. Moreover, health care accessibility itself is perceived narrowly, as it does not seem to take into account the more complex theories of health care accessibility, as already described in 1978 by Tanahashi or more recently by Levesque [29,30].

Partnerships or participation are perceived mainly as consisting of better care for individual health and accepting responsibility, not just participation in decision making. The main actor in the field of eliminating health inequalities is the majority, which defines the “path” that needs to be shown to the Roma, or what is needed to “stimulate” citizens. Rather than emphasizing autonomy and participation, according to our findings from our analyses of the studied documents, we detect an emphasis on education that will help individuals take responsibility for their own destiny.

The analyzed documents contain several elements of assimilation: they primarily see the responsibility being on the part of the Roma themselves to come closer to the majority in their understanding of health and their actions in relation to illness and health. The proposed strategies and measures are guided mainly by what the majority defines as a priority, not by the problems or needs of the Roma themselves. Roma are portrayed in the documents as not “fitting in” with the rest of society [31]; their group identities and practices are described as things that hinder the achievement of the desired goals in the field of health, and their adaptation is perceived as a necessary acceptance of responsibility. In contrast, changes and measures on the part of society are mentioned much less often.

Although the political discourse concerning Roma has shifted more towards human rights, equity and combating discrimination in the Czech Republic [4], subtle forms of anti-Gypsyism still seem to be present. Like in Romania, for example, the conceptualization of Roma in the strategy documents associate Roma with undesired behaviours, identity or values. According to Popoviciu and Tileaga, this has gradually changed over time in Romania, however: “ethnic blame was being inconspicuously couched more fully in the language of modern racism, making it harder to spot and also harder to challenge” [8]. In the Czech Republic, according to our findings, the situation seems to be similar. In this regard, it is important to point out that when such discourse (both homogenizing Roma and muting structural conditions and non-Roma agency) is reflected, mainstreamed, and transformed into specific interventions, it hinders empowerment, participatory and bottom-up actions [1].

### 4.1. Strengths and Limitations

Our study provides a comprehensive analysis of policy documents focusing on a topic which, at least in the Czech Republic, seems to be understudied. It provides evidence of problematic conceptions of Roma in policy documents related to social inclusion and health and demonstrates how anti-Gypsyism and ethnic blame is still present in political discourse. However, some limitations need to be mentioned. Our findings are based on an analysis of policy documents and, consequently, describe a specific view of reality, as policies and actual practices or particular experiences may differ. Another limiting factor of our study is its subjectivity, which is inherently present in qualitative research, where it is generally difficult to separate results from the person of the investigator [32]. To enhance the validity of our research, we used the method of triangulation—our research findings were consulted with the stakeholders working in the field who were involved in various project activities, and our conclusions are supported by the findings of our colleagues, who approached the same topics differently, i.e., through field research or interviews with the actors.

### 4.2. Implications

It would be desirable to find a meaningful and healthy balance in the way policymakers use language in relation to Roma. Describing the “problem” in order to adequately target the intervention should be done without generalizing and supporting stereotypes or by using simplified and essentially erroneous arguments and portrayals. On the other hand, not recognizing ethnicity when dealing with social exclusion and/or health would mean ignoring an ethnic principle that plays an important, though not the only, role in these phenomena. When designing policies and interventions, their nature and the content should avoid the tendency to impose a burden of responsibility on the target population without a clear vision of how willing they are and what steps they will undertake to meet its needs and enable and facilitate its participation during the whole process from design, through to implementation and evaluation.

## 5. Conclusions

In the analyzed policy documents related to social inclusion and health in the Czech Republic, the emphasis lies in individual responsibility in relation to health (mentioning, above all, the risk behaviour of individuals as the cause of the poorer health of Roma), while structural conditions and discrimination are mentioned less and, even when they are, the solution is again seen as being at the individual level. The Roma are described in relation to health primarily as people who should be educated. One of the main means of solving health inequalities is “education” and increasing health literacy. It is assumed that work with Roma patients or clients has a certain specificity, so it is important to train health care professionals in how to deal with Roma patients, to educate them on the specifics of the Roma population. Emphasis on the economic benefits of eliminating health inequalities is greater than on citizens’ rights and the importance of inclusion. When “participation” or “empowerment” are mentioned, this is done so vaguely, without specifics, usually in addition to references to completely non-participatory practices. If the necessity for partnership in the sense of greater activity of the Roma themselves is emphasized, this primarily means care of one’s individual health and taking responsibility, not participation in decision making. The majority are the primary actors in the field of eliminating health inequalities, as they define the “path” that Roma need to be shown or determine what is needed to “stimulate” citizens.

## Figures and Tables

**Table 1 ijerph-17-07739-t001:** List of analyzed documents.

Health Plan for the City of Brno 2018–2030
Health Promotion in Excluded Localities—Reducing Health Inequalities
Strategy to Combat Social Exclusion 2016–2020
Strategy for Integration of the Roma Community in the Central Bohemian Region for 2017–2021
Strategy for Integration of Socially Excluded Localities in the Vysočina Region
Strategy for Integration of the Roma Community in the Moravian–Silesian Region for 2015–2020
Strategy for Health Promotion and Health Services Development in the Ústí nad Labem Region 2015–2020
Strategy for Roma Integration in the South Moravian Region (2014–2018)
Strategy for Roma Integration to 2020
Strategy for Social Integration 2014–2020
Health 2020—National Strategy for Health Protection and Promotion and Disease Prevention
Health 2020. Health Programme for the Vysočina Region
Health 2030: Strategic Framework for Developing Health Care in the Czech Republic to 2030
Health Versus Social Exclusion; Medical-Social Assistance
Health; Health Inequalities in Health Care Accessibility
Report on the Status of the Roma Minority for 2018

**Table 2 ijerph-17-07739-t002:** Categories and codes characteristic for the way Roma are presented in relation to their health in the analyzed documents.

Domain 1. Why address health inequalities at all? “Leave no one behind”
Codes: housing and health, health care efficiency, health inequalities, poverty and health, inclusive concept of health, benefits of improving population health, exclusion as a lack of resources, risks in general—part of a life spent in illness, risks in general—increasing costs, equality, Roma health research
Domain 2. Health as an individual choice: “Let’s live a healthy life”
Codes: poverty and health awareness, education, health promotion as education of patients, prevention, health policing methods in health protection, health literacy, lifestyle of people from excluded locations, lifestyle of Roma as a cause of their poorer health
Domain 3. Structural causes of the poorer health status of Roma: “An unsuitable environment for spreading any education”
Codes: health care accessibility, comprehensive solutions for the structural causes of poorer health, discrimination in access to health care, structurally affected regions, structural causes of exclusion and poorer health status of those living in exclusion
Domain 4. Education of health care professionals on the specifics of the Roma population: “Ignorance causes misunderstanding”
Codes: education of health care professionals, specific needs of Roma clients, Roma participation
Domain 5. Participation primarily as better care of individual health: “The need to stimulate citizens towards responsibility”
Codes: empowerment, Roma participation, partnership, health care professionals as the path to strengthening the role of patients

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
