# Peer review of "Conceptualization of Roma in Policy Documents Related to Social Inclusion and Health in the Czech Republic"

_ijerph, 2020, doi:10.3390/ijerph17217739_

Round 1

Reviewer 1 Report

This article is interesting and relevant to multiple academic fields. I would suggest that the paper has been completed using a robust methodological approach that has facilitated the authors' presentation of the research results and their overarching analysis. This means that firstly the paper has identified how Roma health is potentially negatively impacted by the failure of Czech policy to acknowledge the agency of Roma themselves and the systemic and structural preconditions of their social and economic exclusion that results in their lack of access to and failure to engage with health services. Secondly, the paper has provided its evidence of problematic conceptions of Roma in health policy in the Czech Republic by unpacking those conceptions in a thoughtful analytic framework. In doing so the paper has set out an analytic framework that is applicable to other policy environments and in other jurisdictions. Thus, the paper overall has provided scrutiny of health policy in the Czech Republic in relation to Roma that explains why Roma health remains poor despite ideological overarching policy approaches aligned with human rights discourse. Further, many of the key issues raised in the paper via its analytic framework may be used elsewhere to appreciate Roma exclusion, particularly in relation to responsibilisation, homogenisation and participatory engagement of Roma.  

Author Response

Thank you very much for your positive comment.

The whole text including revised parts was proofread by a professional translator and native speaker.

Reviewer 2 Report

Review of “Conceptualization of Roma in Policy Documents…”

This is an interesting and potentially valuable research project that seeks to understand how the Czech Republic policy makers view health disparities and inclusion as these are directed toward Roma communities. The author(s) conduct a documentary analysis using ethnographic software to expose and analyze common themes in these documents. The results suggest that Roma are described in homogeneous terms and that the focus of documents promoting the reduction of healthcare disparities is focused on changing the individual behaviors of the target group vs. addressing structural inequalities.

There are several things to like here. There’s a clear focus on a set of documents that others can potentially use, increasing the overall utility of the research. The themes make intuitive and analytical sense and they are summarized nicely in a table. The conclusions also make sense (as far as they go, more on that below).

What’s lacking here is two things – (1) broader background on the relation of the Roma to the Czech government historically, and (2) more information on the health behaviors and activities of the group in question (i.e. Roma).

Some broader background is needed here to understand what the history of interactions are between Roma, the Czech government, and the healthcare system generally. We also need more information on how the healthcare system actually works in comparison to government claims about how it works. A healthcare system that fails significant numbers of a native population is not going to treat minorities well and will treat small minorities really poorly (at 2 percent of the population it is safe to say the Roma are a small minority here). The author(s) imply that the overall system needs work but don’t really lay out any specifics here. The competing claims about how the healthcare system works are probably tied up in claims about how minorities should be treated or integrated. A second, somewhat related problem revolves around the author(s) claim that the Roma are themselves diverse and treating them as a single group is problematic. A more detailed background on the relationship between the Roma and the Czech government would provide a basis for the author(s) description of how Roma vary in and of themselves.

The answers to (1) lead to the information needed in (2). What exactly are the health behaviors of the Roma themselves? Do they, in fact, have bad diets? Smoke too much? Use drugs? Drink too much? These may be “Stereotypes” but if people actually engage in these behaviors then they are actually harming their health. If the rates of these behaviors by Roma are no larger than the general population, then it’s a non-factual stereotype. If there are actual differences there, then directing a healthcare system toward changing those is completely justified and the only question is how to do that. The government may be promoting the wrong strategy then (trying to curb “irresponsible behavior”) but the basic facts are not in dispute.

Unfortunately, the Roma are a small enough minority that there is an overwhelming temptation to engage in the individualistic response the Czech government has chosen even if the chances of it working are low. The stakes (for majority groups and the government) simply aren’t high enough to engage in more extensive interventions. If the Roma comprised (say) 15 percent of the population, then there would be some incentive to try something more extensive.

The final thing the paper could use is a more policy-focused alternative. If the government’s focus is wrong, what’s the right focus? This would add considerable value to the paper.

Author Response

Reviewer 2:

R2.1

This is an interesting and potentially valuable research project that seeks to understand how the Czech Republic policy makers view health disparities and inclusion as these are directed toward Roma communities. The author(s) conduct a documentary analysis using ethnographic software to expose and analyze common themes in these documents. The results suggest that Roma are described in homogeneous terms and that the focus of documents promoting the reduction of healthcare disparities is focused on changing the individual behaviors of the target group vs. addressing structural inequalities.

There are several things to like here. There’s a clear focus on a set of documents that others can potentially use, increasing the overall utility of the research. The themes make intuitive and analytical sense and they are summarized nicely in a table. The conclusions also make sense (as far as they go, more on that below).

Response: Thank you for these encouraging words.

R.2.2

What’s lacking here is two things – (1) broader background on the relation of the Roma to the Czech government historically, and (2) more information on the health behaviors and activities of the group in question (i.e. Roma).

Some broader background is needed here to understand what the history of interactions are between Roma, the Czech government, and the healthcare system generally. We also need more information on how the healthcare system actually works in comparison to government claims about how it works. A healthcare system that fails significant numbers of a native population is not going to treat minorities well and will treat small minorities really poorly (at 2 percent of the population it is safe to say the Roma are a small minority here). The author(s) imply that the overall system needs work but don’t really lay out any specifics here. The competing claims about how the healthcare system works are probably tied up in claims about how minorities should be treated or integrated. A second, somewhat related problem revolves around the author(s) claim that the Roma are themselves diverse and treating them as a single group is problematic. A more detailed background on the relationship between the Roma and the Czech government would provide a basis for the author(s) description of how Roma vary in and of themselves.

Response: Thank you for your comment. We agree that some more insight regarding issues you proposed would be valuable. Also, we agree that overall performance of health systems is crucial and it determines further how it will treat minorities.  We have added following information about these issues in the Introduction (lines 67-92).  

Introduction:

“The Czech Republic is among the EU Member States with the largest Roma communities and facing the most acute challenges [9]. The Roma population in the Czech Republic is estimated to be 240,300 (2.2% of the population) [10], and about 50% of Roma live in social exclusion [11]. Roma are underrepresented when it comes to political participation and representation. According to the Civil Society Monitoring Report on Implementation of the National Roma Integration Strategies, the representation of Roma men and women in the Czech Parliament after elections in 2017 was zero, and ten Roma men and women were working in the state administration. Moreover, the Czech political scene is marked by the presence of openly anti-Roma, racist, xenophobic parties, and anti-Roma or racist speech appears abundantly from standard, mainstream parties and politicians [9].

Regarding the health care system of the Czech Republic, its overall performance may be crucial for determining how it acts towards Roma. Overall, in the past decades, the health care system has been successful in bettering its performance in various routinely monitored indicators, such as health expenditures per person, effectiveness, accessibility and resilience. There seem to be only marginal unmet needs for medical care, which can be attributed to universal population coverage, a broad benefits basket, low cost-sharing and a high density of providers [12]. However, despite many improvements, the Czech health care system remains below the EU average in many other indicators [12]. The Ministry of Health itself acknowledges significant disparities in the fulfilment of the objectives of individual action programmes of Health 2020 and the largely delayed implementation of the strategy due to insufficient financial coverage for the planned activities [13]. The health status of the Czech population is characterized by substantial inequalities by education and income ([12, 14]). Vast regional disparities are characteristic of the overall distribution of primary care providers, who play a vital role in health promotion and prevention and as the first contact point in the health care system across the country [12]. The planning of health care services is aggravated by fragmented responsibilities in planning among several stakeholders (health insurance funds, self-governing regions, the Ministry of Health, the Ministry of Social Affairs, etc.). This causes the higher health needs in some Czech regions to not be properly accounted for in planning [12].”

Added sources:

  • European Commission. Civil society monitoring report on implementation of the national Roma integration strategies in Czech Republic. 2018. Available online: https://cps.ceu.edu/sites/cps.ceu.edu/files/attachment/basicpage/3034/rcm-civil-society-monitoring-report-1-czech-republic-2017-eprint-fin.pdf (accessed on 18 October 2020)
  • OECD/European Observatory on Health Systems and Policies. Czechia: Country Health Profile 2019, State of Health in the EU. 2019. OECD Publishing: Paris, France.
  • Ministry of Health of the Czech Republic. Informace o stavu realizace Zdraví 2020 – Národní strategie ochrany a podpory zdraví a prevence nemocí za období říjen 2017 – září 2018. Available online: https://www.mzcr.cz/informace-o-stavu-realizace-zdravi-2020-narodni-strategie-ochrany-a-podpory-zdravi-a-prevence-nemoci-za-obdobi-rijen-2017-zari-2018/ (accessed on 18 October 2020)
  • Bosakova, L.; Rosicova, K.; Filakovska Bobakova, D.; Rosic, M.; Dzurova, D.; Pikhart, H.; Lustigova, M.; Santana, P. Mortality in the Visegrad countries from the perspective of socioeconomic inequalities. J. Public Health 2019, 64, 365-376. doi: 10.1007/s00038-018-1183-6

R.2.3

The answers to (1) lead to the information needed in (2). What exactly are the health behaviors of the Roma themselves? Do they, in fact, have bad diets? Smoke too much? Use drugs? Drink too much? These may be “Stereotypes” but if people actually engage in these behaviors then they are actually harming their health. If the rates of these behaviors by Roma are no larger than the general population, then it’s a non-factual stereotype. If there are actual differences there, then directing a healthcare system toward changing those is completely justified and the only question is how to do that. The government may be promoting the wrong strategy then (trying to curb “irresponsible behavior”) but the basic facts are not in dispute. A few corrections are needed to the English.

Response: Thank you for your comment. We agree that targeting health endangering behaviours in this particular population is accurate as the differences truly exist. We also agree that the way how policy makers approach this particular issue is important, particularly because such behaviours should be viewed more as the result of unhealthy socialisation, adaptation and coping strategies which are natural reaction on social exclusion, disadvantage, stress and oppression than a matter of personal choice. We have added following in the Introduction (lines 93 – 110) and Discussion (643 – 648):

Introduction:  

“As the evidence suggests, the health status of Roma is worse than that of the non-Roma population in the Czech Republic. Roma life expectancy is about 10–15 years less than the majority population and there is a significantly higher prevalence of chronic illnesses in the Roma population. The infant mortality rate is also higher among Roma [15]. The Czech Roma population smokes significantly more than the non-Roma population and only low numbers of Roma (adults and/or children) undertake physical activities to stay healthy. The Czech Roma population typically has a poor diet, with less than the recommended daily intake of vegetables, fruit and dairy products, and above the recommended intake of meat, sweets and processed food [15]. The reasons for the worse health status of Roma are complex and cannot be easily identified merely as an “unhealthy lifestyle” or “risky health behaviour” or seen as individual characteristics, since health-related behaviour is rather the consequence of exclusion, worse living conditions (especially housing), poverty and the lower social, economic and cultural capital of the Roma population [15, 16]. What is more, the European Roma population generally faces disproportionate barriers to accessing health services, and the Czech Roma population is no exception. This situation is caused by several factors: living in remote areas without necessary health and transport infrastructure, a lack of financial sources or insurance, as well as discriminatory attitudes [15, 17, 18]. The health needs of the Roma population are often invisible because of the absence of research but also due to the absence of advocacy on their behalf, as Koupilová et al. suggest [19].”

Discussion:

“Although targeting health-endangering behaviours seem to be reasonable given that the Roma – particularly those living in social exclusion – do engage in such behaviours more frequently [15], the approach and rationale for actions as found in the policy documents seem to lack the awareness of the root causes of engaging in such behaviours, specifically in the population of socially excluded Roma [24] on one hand or the theory of planned behaviour [25] on the other.”

Added sources:

  • European Commission. Roma Health Report – Health Status of the Roma Population. Data Collection in the Member States of the European Union. 2014. Available online: https://ec.europa.eu/health/sites/health/files/social_determinants/docs/2014_roma_health_report_en.pdf (accessed on 18 October 2020)
  • Kajanová, A. Sociální determinanty zdraví vybraných romských komunit. Dissertation Thesis, Jihočeská univerzita v Českých Budějovicích, České Budějovice, 2009.
  • Davidová, E. Kvalita života a sociální determinant zdraví u Romů v České a Slovenské republice. Triton: Praha, Czech Republic, 2010.
  • Koupilová, I.; Epstein, H., Holcík, J.; Hajioff, S.; McKee, M. Health needs of the Roma population in the Czech and Slovak Republics. Sci. Med. 2001, 53, 1191-1204. doi: 10.1016/s0277-9536(00)00419-6
  • Belak, A. The health of segregated Roma: first-line views and practices: a case study in Slovakia using ethnographic methods. Dissertation Thesis, University of Groningen, Groningen, 2019.
  • Ajzen, I. The theory of planned behavior. Behav. Hum. Decis. Process. 1991, 50, 179-211. doi:10.1016/0749-5978(91)90020-T

R.2.4

Unfortunately, the Roma are a small enough minority that there is an overwhelming temptation to engage in the individualistic response the Czech government has chosen even if the chances of it working are low. The stakes (for majority groups and the government) simply aren’t high enough to engage in more extensive interventions. If the Roma comprised (say) 15 percent of the population, then there would be some incentive to try something more extensive.

Response: Thank you for raising this issue. The Czech Republic has committed itself to complying with various conventions and strategies. Thus despite the fact that 2,2% of the whole population might be viewed as not worth to invest too much, the fact is that a lot of effort and investments are made to tackle health inequities faced by Roma as these are one of the largest in the word. However, the success and effect of these efforts are questionable precisely because participatory and bottom up approaches are mostly mentioned only on the paper but not really implemented. Evidence regarding effectiveness of public health interventions is very limited thus it is hard to make any conclusions on this. Although this is an important issue which surely deserves attention, it is far beyond the scope of our article.

R.2.5

The final thing the paper could use is a more policy-focused alternative. If the government’s focus is wrong, what’s the right focus? This would add considerable value to the paper.

Response: Thank you for this suggestion. We enriched the discussion as follows (lines 724 – 734):

4.2. Implications

 “It would be desirable to find a meaningful and healthy balance in the way policymakers use language in relation to Roma. Describing the “problem” in order to adequately target the intervention should be done without generalizing and supporting stereotypes or by using simplified and essentially erroneous arguments and portrayals. On the other hand, not recognizing ethnicity when dealing with social exclusion and/or health would mean ignoring an ethnic principle that plays an important, though not the only, role in these phenomena. When designing policies and interventions, their nature and the content should avoid the tendency to impose a burden of responsibility on the target population without a clear vision of how willing they are and what steps they will undertake to meet its needs and enable and facilitate its participation during the whole process from design, through implementation to evaluation.”  

Reviewer 3 Report

This paper aims to examine the conceptualization of Roma in policy documents related to social inclusion and health in the Czech Republic. The topic is interesting  and will bring new information to health care policy. Many minor suggestions need to clarify in the content:

1.method part described this project included a qualitative analysis related to health inequalities or the social inclusion of Roma. However, the title of this paper is social inclusion and health. Please make sure the correct research title and purposes. 

2. How to make sure the "Table 1. List of analysed documents" is enough data or materials to this study? and there are too many issues in each document to narrow down the issues.

3.the three main research questions seem too broad to generalize the finding, and how to analyse the document by people or software?

4.the results should present by sub-theme in each theme, to describe the results appropriately.

5. the author should describe the study limitations in the content.

Author Response

Reviewer 3:

R3.1

Method part described this project included a qualitative analysis related to health inequalities or the social inclusion of Roma. However, the title of this paper is social inclusion and health. Please make sure the correct research title and purposes. 

Response: Thank you for pointing out this inconsistency. In order to make research title and purposes more consistent, we clarified the purposes of analysis (lines 131-132, subsection 2. 1. Design) as follows:

“The project activities also included a qualitative analysis of documents related to social inclusion and health.”

R3.2

How to make sure the "Table 1. List of analysed documents" is enough data or materials to this study? and there are too many issues in each document to narrow down the issues.

Response: Thank you for your comment. We provided more details on the process of sampling (lines 141 – 146, subsection 2. 2. Sample).

“The list of the documents for analysis was consulted and finalized in cooperation with several stakeholders working in the field of social inclusion or health. Besides the recommendation of the stakeholders, the selection of documents was led by the principle of theoretical saturation, i.e. it ended at the moment when no new or relevant data seem to emerge by analysing new documents [22].”

Added source:

  • Strauss, A.; Corbin, J. Basics of Qualitative Research: Techniques and procedures for developing grounded theory, 2nd ed.; Sage Publications: Thousand Oaks, United States, 1998.

R3.3

The three main research questions seem too broad to generalize the finding, and how to analyse the document by people or software?

Response: Thank you for your comment. We provided more details on our analytical procedure (lines 153 – 159, subsection 5. 3. Procedure). The information on the analysis by software was included in the original part of manuscript (line 159).

“The original research questions were, in accordance with the inductive logic of qualitative analysis, rather broad [22] and the analysis became more focused with the first topics and categories emerging in the data, as described below.”

Added source:

  • Strauss, A.; Corbin, J. Basics of Qualitative Research: Techniques and procedures for developing grounded theory, 2nd ed.; Sage Publications: Thousand Oaks, United States, 1998.

R3.4

The results should present by sub-theme in each theme, to describe the results appropriately.

Response: Thank you for your comment. We clarified the status of our categories and sub-themes (codes) within analytical process and data presentation (lines 161 – 165, subsection 2. 4. Statistical analysis and reporting and lines 171 – 172, subsection 3. Results).

“For the analysis, the most important particular codes were then assigned into broader categories which reflect the individual subchapters of the analysis below. Proceeding from the particular to the more general is one of the key principles of the qualitative analysis [22] and enables us to present our findings as contextualized.” 

“Table no. 2 presents the names of these categories/domains and codes (as the tools for reducing, sorting and categorizing data) that are included in them.”

Added source:

  • Strauss, A.; Corbin, J. Basics of Qualitative Research: Techniques and procedures for developing grounded theory, 2nd ed.; Sage Publications: Thousand Oaks, United States, 1998.

R3.4

The author should describe the study limitations in the content.

Response

Thank you for your comment. We added the subsection 4. 1. Strengths and Limitations (lines 711 – 723).

4.1. Strengths and Limitations

“Our study provides a comprehensive analysis of policy documents focusing on a topic which, at least in the Czech Republic, seems to be understudied. It provides evidence of problematic conceptions of Roma in policy documents related to social inclusion and health and demonstrates how anti-Gypsyism and ethnic blame is still present in political discourse. However, some limitations need to be mentioned. Our findings are based on an analysis of policy documents and, consequently, describe a specific view of reality, as policies and actual practises or particular experiences may differ. Another limiting factor of our study is its subjectivity, which is inherently present in qualitative research, where it is generally difficult to separate results from the person of the investigator [32]. To enhance the validity of our research, we used the method of triangulation – our research findings were consulted with the stakeholders working in the field who were involved in various project activities, and our conclusions are supported by the findings of our colleagues, who approached the same topics differently, i.e. through the field research or interviews with the actors.”

Added source:

  • Aldridge, D.; Aldridge, G. A personal construct methodology for validating subjectivity in qualitative research. Arts Psychother. 1996, 23, 225-236. doi:10.1016/0197-4556(96)00023-8

Reviewer 4 Report

The aim of this paper is to examine the conceptualization of Roma in policy documents related to social inclusion and health in the Czech Republic and to know how are the Roma presented in these documents as actors in relation to health, what role is ascribed to them and how much is their participation expected.

Results highlight that Roma are described in relation to health primarily as people who should be educated, and health inequalities or structural discrimination is vaguely mentioned. At the same time, the study demonstrates that, although the political discourse concerning Roma has shifted more towards human rights, equity and combating discrimination in the Czech Republic, subtle forms of anti-Gypsyism still seem to be present.
On one hand, the introduction provide sufficient background and include relevant references. The research design is appropriate and the methods are adequately described. The results are clearly presented and the conclusions are supported by the results and the theoretical background used.

On the other hand, this paper has a high significance of content and could be an interesting paper for the readers, especially because demonstrates how anti-Gypsyism is still present in all society spheres, as in the public administration.

As a researcher and as a Roma, I would like to thank authors of this study to develop this kind of researches, in order to fight against anti-Gypsyism and help Roma community to be well represented in public documents. This study could be useful to develop and implemented Roma strategies that promote, truly, equality, respect and inclusion for Roma.

Author Response

Thank you very much for this supportive and stimulating comment. 

The whole text including revised parts was proofread by a professional translator and native speaker.